

**Shallow groundwater in sub-Saharan Africa: neglected opportunity**
**for sustainable intensification of small-scale agriculture?**
John Gowing[1], Geoff Parkin[2], Nathan Forsythe[2], David Walker[2], Alemseged Tamiru
Haile[3], Demis Alamirew[4]
1. School of Agriculture, Food & Rural Development, Newcastle University, Newcastle upon
Tyne, NE1 7RU, United Kingdom
2. School of Civil Engineering & Geosciences, Newcastle University, Newcastle upon Tyne,
NE1 7RU, United Kingdom
3. International Water Management Institute, P.O. Box 5689, Addis Ababa, Ethiopia.
4. Geological Survey of Ethiopia, P. O. Box 30912, Addis Ababa, Ethiopia
Correspondence to: john.gowing@ncl.ac.uk



## 14 **Abstract**

There is a need for an evidence-based approach to identify how best to support development
of groundwater for small scale irrigation in sub-Saharan Africa (SSA). We argue that it is
important to focus this effort on shallow groundwater resources which are most likely to be
used by poor rural communities in SSA. However, it is important to consider constraints,
since shallow groundwater resources are likely to be vulnerable to over-exploitation and
climatic variability. We examine here the opportunities and constraints and draw upon
evidence from Ethiopia. We present a methodology for assessing and interpreting available
shallow groundwater resources and argue that participatory monitoring of local water
resources is desirable and feasible. We consider possible models for developing distributed
small-scale irrigation and assess its technical feasibility. Because of power limits on water
lifting and also because of available technology for well construction, groundwater at depths
of 50m or 60m cannot be regarded as easily accessible for small-scale irrigation. We
therefore adopt a working definition of shallow groundwater as $< 20$ m depth.
This detailed case study in the Dangila woreda in Ethiopia explores the feasibility of
exploiting shallow groundwater for small-scale irrigation over a range of rainfall conditions.
Variability of rainfall over the study period (9% to 96% probability of non-exceedance) does
not translate into equivalent variability in groundwater levels and river baseflow.
Groundwater levels, monitored by local communities, persist into the dry season to at least
the end of December in most shallow wells, indicating that groundwater is available for
irrigation use after the cessation of the wet season. Arguments historically put forward
against the promotion of groundwater use for agriculture in SSA on the basis that aquifers are
unproductive and irrigation will have unacceptable impacts on wetlands and other
groundwater-dependent ecosystems appear exaggerated. It would be unwise to generalise
from this case study to the whole of SSA, but useful insights into the wider issues are
revealed by the case study approach. We believe there is a case for arguing that shallow
groundwater in sub-Saharan Africa represents a neglected opportunity for sustainable
intensification of small-scale agriculture.






## 1. Introduction

### 1.1 Context

There is abundant groundwater in Africa; more than 100 times the annual renewable freshwater resource and 20 times the amount of freshwater stored in lakes (MacDonald et al., 2012), but its productive use for irrigation in sub-Saharan Africa (SSA) remains low. Examining the evidence on use of groundwater for irrigation in SSA, Pavelic et al. (2013) argued for action to unlock its potential for improving livelihoods of smallholder farmers. We examine here the opportunities and constraints and draw upon evidence from Ethiopia to demonstrate the case for action to promote use of *shallow* groundwater, in particular, for small-scale irrigation in SSA.

Historically, groundwater exploitation has not been seen as an important component of water resources development in SSA (Braune and Xu, 2010). Its contribution to rural water supply is recognised, but groundwater has been seen more as a local resource, which supports domestic demand, rather than as a strategic resource which can support productive use and economic development. Arguments historically put forward against the promotion of groundwater use for agriculture in SSA include that aquifers are said to be low in transmissivity and that well yields are inadequate to support agricultural development at scales larger than garden irrigation, particularly in the weathered crystalline basement rocks that extend over about 40% of the African land mass (Wright, 1992; Chilton & Foster, 1995; MacDonald et al. 2012). It has also been argued that groundwater use for irrigation will have unacceptable impacts on wetlands and other groundwater-dependent ecosystems and on domestic supplies (Adams, 1993; Giordano & Villholth, 2007; MacDonald et al., 2009).

However, the agenda has shifted and groundwater irrigation (GWI) by smallholder farmers is increasingly being promoted by governments, donors and NGOs (Abric et al., 2011; CAADP, 2009; Chokkakula and Giordano, 2013). GWI is now seen as an important vehicle to promote poverty alleviation, food security, rural employment, market-oriented agriculture and climate change adaptation (Ngigi, 2009). Groundwater resources are ideally suited to development of 'distributed irrigation systems' (Burney et al., 2013) in which farmers enjoy far greater autonomy and flexibility of water supply than is possible through canal systems.



Survey evidence shows that smallholder farmers prefer GWI (Abric et al., 2011; Giordano et
al., 2012; Villholth, 2013).

The global area equipped for irrigation has been estimated (Siebert et al., 2010) as 301 Mha,
of which 38% depends on groundwater. In SSA the extent of GWI is much less with only 6%
of the irrigated area reported by Siebert et al. (2010) and 10% by Giordano (2006) to be
supported by groundwater. However, a note of caution is necessary when considering official
statistics because of problems of definition and invisibility of so-called 'informal irrigation'
(Giordano, 2006; Frenken, 2005). Using evidence from various countries in SSA, Villholth
(2013) revised this estimate to 20% of the total irrigated area. Notable examples of public
sector initiatives exist, such as in the Fadama Development Programme in Nigeria (Abric et
al., 2011), but it is important to recognise the dominance of the informal sector, which is
characterised by autonomous farmer initiatives based upon the exploitation of shallow
groundwater resources. Such initiatives receive little official recognition and support
(Chokkakula and Giordano, 2013) and there is an urgent need to develop capacity for the
state to function in a dual role as facilitator and regulator of GWI. We argue that it is
important to focus this effort on *shallow* groundwater resources which are most likely to be
used by poorer rural communities in SSA.

1.2     Shallow groundwater: the opportunity
In the past few decades in Asia, a paradigm shift has occurred in irrigation practice, such that
distributed irrigation using privately owned wells and small motorised pumps has expanded
rapidly. This development has enabled smallholder farmers to diversify their farming systems
and grow high-value crops for the market. There is growing, but patchy, evidence that a
similar 'irrigation revolution' is happening in SSA (Dessalegn and Merrey, 2015).

Irrigation does not currently play a major role in African agriculture; the area equipped for
irrigation as a percentage of total cultivated land is 19.4 % globally, but only 3.3% for SSA
(Siebert et al., 2010), where agriculture remains almost entirely rainfed (You et al., 2010).
There have been many assessments of the irrigation potential (eg. Frenken, 2005) and
ambitious plans for its expansion, such as Commission for Africa (2010), which proposed
doubling the area under irrigation. In reviewing the investment needs on behalf of the World
Bank, You et al. (2010) examined biophysical and socio-economic factors affecting large and



small-scale irrigation development. They found that small-scale irrigation offered far greater
potential than large scale; offering five times the expansion potential and double the
estimated rate of economic return. GWI can make an important contribution provided that the
focus is on shallow groundwater using technology that is accessible to small-scale farmers.

A simple typology of GWI is suggested by Villholth (2013) based on two key characteristics:
funding source (ie. private or public) and depth of groundwater (ie. deep or shallow). In most
parts of SSA existing GWI is privately funded and utilises shallow wells. It is used primarily
in high-value, market-oriented production (Shah et al., 2013) and women often play a
prominent role (van Koppen et al., 2013).
1.3    Shallow groundwater: anticipated constraints
Shallow groundwater is accessible to small-scale farmers with simple technologies for well
construction and water lifting and offers the best opportunity to develop low-cost GWI.
However, it is important to consider constraints since shallow groundwater resources are
likely to be vulnerable to over-exploitation and climatic variability. Villholth (2013) notes
that sustainable development of groundwater use for irrigation is limited by "replenishment
rates … extractability in some regions … and as a provider of environmental services", and
argues that there is a need for understanding integrated groundwater and surface water
systems at different scales".

Broad scale assessments of groundwater resource potential at national or continental scales
(e.g. MacDonald et al., 2012) and at sub-national scales (e.g. Awulachew et al., 2010)
provide an indication of the spatial extent and storage volume in aquifer formations, but an
assessment of the resource potential is critically dependent on understanding groundwater
dynamics. A recent review of groundwater conditions in 15 SSA countries concluded that
"information on aquifer characteristics, groundwater recharge rates, flow regimes, quality
controls and use is still rather patchy" (Pavelic et al., 2012b). There is widespread use of
shallow groundwater for domestic supply in most SSA countries, and indigenous knowledge
generally exists on the seasonal performance of wells during typical and drought years.
However, this knowledge is localised, qualitative and unrecorded. In contrast, there is
increasing availability of relevant global remote sensing data including topography, land
cover, soil moisture and climate products providing broad scale information that can be used
to estimate resource availability.




Broad scale quantitative mapping of groundwater potential for Africa was revisited by
Altchenko and Villholth (2015) who considered the potential for sustainable GWI based on
renewable groundwater resources with $0.5^o$ spatial resolution. They adopted an approach
based on conservative estimates of groundwater recharge and alternative scenarios for
allocation of groundwater to satisfy environmental requirements. They concluded that
throughout most of the Sahel and for the eastern tract of SSA from Ethiopia to Zimbabwe
renewable groundwater is under-exploited, and in some countries is sufficient to irrigate all
cropland. Any such assessment is subject to uncertainty and temporal variability of recharge
estimates. Due to the fragmented and localised nature of shallow groundwater resources
(Pavelic et al., 2012a) their capacity to buffer against inter-annual variability is expected to be
less than in the case of extensive deep aquifer formations.

As noted by Edmunds (2012), a major limiting factor is the need to identify whether the
stored groundwater is a renewable or a non-renewable resource, which depends on local
hydrogeological settings as well as regional climate. Therefore, there is a need to improve
understanding of available groundwater resources and to consider likely impacts of future
trends in climate and land use. In order to allow for balanced consideration of the
opportunities for and constraints to GWI from shallow aquifers in SSA, we report here a case
study in Ethiopia. We present a methodology for assessing and interpreting available shallow
groundwater resources and argue that participatory monitoring of local water resources is
desirable and feasible. We consider possible models for developing distributed GWI and
assess its technical feasibility.

**2.    Study area**
The appropriate scale for the case study was considered to be a single administrative district
(known in Ethiopia as a *woreda*) as this allowed consideration of both technical and socio-
economic aspects of groundwater resource assessment and management. In view of the
priority given to agricultural transformation in the area and availability of hydrogeological
data, the Tana basin was selected as a suitable site for the pilot study. Several *woredas* in the
basin were considered on the basis of their accessibility, the dominant farming system and





their status within the agricultural growth strategy. Dangila woreda was selected as the case
study site (Figure 1).
Dangila woreda is situated in the north-western highlands with altitudes generally between
1850m to 2350m. Dangila town is situated along the Addis Ababa-Bahir Dar road at a
distance of 60 km south west of Bahir Dar. Part of Dangila woreda drains north-east towards
Gilgel Abay River and Lake Tana; the remaining area drains either west or south-west
towards Beles River, both of these are part of the Abay (Blue Nile) tributary of the River
Nile. The climate is sub-tropical with annual rainfall around 1600mm and the main rainy
season (known as *Kiremt*) occurring in June-September.
The total population of Dangila woreda is estimated at about 200,000 people in an area of
about 800 km$^2$. Crop–livestock mixed subsistence farming is the primary source of
livelihood. According to a recent survey (Belay and Bewket, 2013) approximately 14% of
cropland is irrigated. This compares with estimates for Ethiopia as a whole of 1.8% by
Siebert et al. (2010) and 2.5% by Altchenko and Villholth (2015). Irrigation is mainly by
means of shared gravity diversions from seasonal and perennial streams, though there are
some reports of water lifting. There are many shallow (up to 12m) dug wells throughout the
woreda, but they are used primarily for domestic supply with only small pockets of garden
irrigation. There are some deeper drilled wells fitted with hand-pumps and some springs have
been developed for community water supply.
Ethiopia's hydrogeology is complex. Basement aquifers, volcanic aquifers and Mesozoic
sediment aquifers are most extensive, but these are generally poor aquifers and consequently,
alluvial and/or Quaternary aquifers are more important. The geology is often highly varied
and, due to tectonic movement, areas with very shallow groundwater can occur alongside rift
areas with very deep groundwater. Kebede (2013) mapped the extent of alluvio-lacustrine
sediments in Ethiopia covering around 25% of the total land area. The alluvial deposits are of
two types: (1) extensive alluvial plains and (2) more localised strips of land and river beds
along rivers and streams occurring in most places both in the highlands and in the lowlands.
Existing mapping of shallow aquifers shows an extensive area of shallow regoliths to the
south of Lake Tana. The study site was selected to allow its exploration as a representative
shallow aquifer formation.






**Figure 1 here**


At the case study site the geology consists of predominantly Quaternary basalt and trachyte
above Eocene Oligocene basalts and trachyte: the ages of these formations are taken from the
1:2,000,000 scale Geological Map of Ethiopia (Tefera et al., 1996). Outcrops are visible in
river beds and occasionally on steeper slopes and in a few man-made excavations. The
basalts are variously massive, fractured and vesicular with variations occurring over short
distances. The more massive basalt generally forms higher ground, with valleys and
floodplains overlying more fractured and vesicular basalt which is more easily weathered and
eroded. Above the solid geology lies weathered basalt regolith, itself overlain by red soils.
The red soils become more lithic and clayey with depth, grading into the regolith usually with
no obvious boundary. The regolith becomes greyer and stronger and has to be chiselled as it
deepens, though it is still quite friable. The most friable regolith is the result of weathering of
low-density vesicular basalt.
The superficial materials underlying the floodplains are often browner in colour, being more
organic-rich. Deep and wide desiccation cracks suggest a high clay content, though these
alluvial materials are occasionally very sandy and gravelly. The depth to the top of the solid
geology is highly variable. Wells are typically excavated until further excavation becomes
impossible, therefore the location of the rock-head can be inferred from well depth. The
rivers have often incised to the level of the rock-head, where solid basalt forms the river bed
with banks of only 1 to 3 m in height.


**3. Feasibility of irrigation from shallow groundwater**
Previous studies have estimated the extent of groundwater irrigation potential across SSA,
and most recently, Altchenko and Villholth (2015) identified the scope for developing small-
scale GWI. They concluded that the semi-arid Sahel and East Africa regions offer appreciable
potential. In Ethiopia, their estimate of sustainable GWI potential based on renewable
groundwater was in the range $1.8 \times 10^6$ to $4.3 \times 10^6$ hectares (depending on provision for
environmental requirements). This represents at least a ten-fold increase on the current extent
of $117 \times 10^3$ hectares as estimated by Villholth (2013). However, assessment of potential





based only on estimated recharge does not provide a reliable indication of the scope for future
expansion, which may be constrained by restrictions on access to the resource. The case study
site provides an opportunity to explore these constraints through a feasibility assessment.
Assessing technical feasibility of small-scale GWI involves balancing considerations of
water-table depth, well yield, technology (power) available for pumping, crop water demand
and area irrigated.
3.1    Depth to groundwater
Most of the literature on groundwater in SSA considers 'shallow' groundwater as any aquifer
up to 50 m or 60 m depth (Pavelic et al., 2012a). However, much of the existing small-scale
GWI depends on a water-table depth less than 5m. Because of power limits on water lifting
and also because of available technology for well construction, groundwater at depths of 50m
or 60m cannot be regarded as easily accessible for small-scale irrigation. We therefore
adopted a working definition of <20 m depth as also adopted by Villholth (2013).
Woldearegay and van Steenbergen (2015) adopted a working definition of <30 m depth for
shallow dug wells in northern Ethiopia.
3.2    Well yield
Typical well yields are reported (MacDonald et al., 2012; Pavelic et al., 2012a) as $1 - 5$ l/s for
volcanic and consolidated sedimentary aquifers. Crystalline basement rocks have lower
yields, generally less than 0.5 l/s, though a significant minority of areas have yields that are in
excess of 1 l/s. There is a clear tendency for groundwater development to focus on deeper
aquifers with higher well yields. In northern Ethiopia, Woldearegay and van Steenbergen
(2015) reported that drilled wells constructed to 80 m or deeper were found to be highly
productive (well yield $> 3$ l/s). However, they also reported that many of these wells were not
operational and many were damaged. There is an apparent conflict between resource
potential and resource access and it is important to consider the constraint imposed by water
lifting technology.
3.3    Water lifting technology
Currently available options are rope and bucket (human power), treadle pump (human
power), chain-and-washer pump (human power), small centrifugal pumps (petrol or diesel
power), submersible pumps (solar power). Important considerations are (a) power available
for lifting water and (b) limit on suction lift.




In the case of human power, a reasonably fit human can sustain a power output of 75W
(Fraenkel, 1986). The type of water lifting device makes little difference to power
requirement, but does affect ability to sustain it for long periods. The pumping rates which
can be achieved assuming a water lifting device with 50% efficiency are shown in Table 1. In
the case of animal power, capabilities of draft animals vary (Fraenkel, 1986). Assuming again
50% efficiency, Table 2 shows the pumping rates which can be achieved for various animals.

**Table 1 here**

**Table 2 here**
Small motorised pumps with rated power output of 0.5hp (375W) or 1hp (750W) are most
likely to be appropriate for petrol/diesel powered pumping from shallow wells. Costs are
currently around \$250. Assuming 50% efficiency, it can be seen that pumping rates will be in
the same range shown above for animal power. However, it should be noted that actual
operating efficiency may be lower (perhaps 25%) for commonly available centrifugal pumps
because of the nature of the efficiency curve for such pumps.

The issue of limit on suction lift applies to any rotodynamic pumps (centrifugal or axial
flow). For such pumps the theoretical limit to suction lift is around 10m but the practical limit
is more like 7m where the pump is installed at sea level. Given that many applications in SSA
may be at altitudes up to 2000m, the limit on suction lift may be as little as 3m. Clearly this is
an important consideration for pumping from a well. A pump installed at the surface can be
used for only very shallow water-table conditions (say 3-5m depth). It may be possible to
modify well design to allow for the pump to be installed on a platform at an intermediate
depth, but practical considerations will still limit applications to water-table depths not
exceeding 10m, and this also represents a risk of aquifer pollution.

To avoid the suction lift constraint, alternative types of pump are required. Handpumps
installed on typical water supply wells are positive displacement (piston and valve type)
pumps. The Rower pump (Fraenkel, 1986) is a piston pump developed for irrigation use
which can deliver around 2.7 m$^3$/h for a lift of 5-6m, which corresponds to the pumping rate
calculated above. The treadle pump (Kay and Brabben, 2000) is a reciprocating diaphragm
pump developed for irrigation use for which quoted delivery rate is again around 3m$^3$/h for a





lift of around 5m. Cost is comparable to a small motor pump at around $250. The main
difference between various types of hand pump appears to be mainly ergonomic such that the
ability to sustain pumping for extended periods may vary but rate of pumping stays much the
same. Motorised positive displacement pumps exist that could be used in principle but this
requires a long drive-shaft to deliver power from a motor on the surface. The alternative is to
use a submersible pump which uses an electric motor which is integral with the pump, both
being installed below the water-table. Availability of electrical supply to the well is an
obvious constraint on electric submersible pumps but solar power is becoming a feasible and
affordable option (Burney et al., 2010).

Matching the rate of pumping to well yield is another consideration in order to avoid
pumping the well dry. It will be seen that a well yield of 1 l/s does not represent a constraint
to human power water lifting but does become a problem with mechanically powered
pumping. Large diameter dug wells provide buffer storage which reduces the problem.

3.4   Crop water demand
Irrigation demand depends on crop type and local environmental conditions, but these do not
make a big difference when considering general feasibility. For the range of crops and
conditions likely to be encountered at typical GWI sites, a crop water demand of 5-8mm/day
can be assumed. Distance of delivery from the well to the crop will be short, so it is
reasonable to assume an irrigation efficiency of 80%. Under these assumptions, daily water
use (m³/day) can be calculated as shown in Table 3.
**Table 3 here**
It is apparent that human powered water lifting cannot irrigate more than 0.1ha for a water-
table deeper than about 3m. For a water-table at 10m depth it requires 3 to 4 hours continuous
effort to irrigate an area of 0.1ha. This is consistent with expected limit on total human power
input of 250 to 300 Wh per day (Fraenkel, 1986). Animal power will allow an increase in the
area of irrigation to about 0.5ha. However the associated rate of pumping may exceed
expected well yield and the system may actually be limited by the aquifer rather than by
power for water lifting.





Motorised pumps at 0.5hp (375W) deliver a flowrate very similar to what is achievable with
animal power and the same considerations therefore apply. However, long duration
continuous pumping is achievable, and it is feasible to irrigate up to 1 hectare from a single
well pumping from 20m deep. Motorised pumps at 1hp (750W) deliver a flowrate that is
above the expected yield from shallow aquifers. Continuous pumping from the well will
therefore not be possible in many cases. It will be desirable to adopt a well design that
increases yield (galleries) or provides storage (over-size well). In most cases there will be no
advantage in adopting a motorised pump rated at more than 0.5 hp (375 W).

A well yield of 3.6 m$^3$/h is equivalent to continuous pumping at 1 l/s, which is a low rate for
efficient irrigation. Pumping to an above-ground storage tank will offer an improved system.
Modular drip irrigation kits (Burney et al., 2013) can overcome this limitation.


**4.     Assessment of the shallow groundwater resource**
4.1     Methodology
*Hydrogeological assessment*
Hydrogeological assessments of the Dangila woreda were conducted between October 2013
and November 2015. The pre-existing geological map was reinterpreted on the basis of
observation of surface features combined with geophysical investigations and sampling from
dug wells and springs. Evaluation of the controlling factors for groundwater movement and
storage, and identification of geological structures (faults, lineaments, joints) and their role to
control flow direction in relation to the direction of major and minor structures was evidenced
by measurement or estimation of spring discharge, estimation of dug well yield based on
users' information, and measurement of some stream flows. Rivers were walked in order to
accurately locate (using a GPS) perennial and seasonal reaches, and water depth, channel
incision and bank width was measured while geology of the river banks and river bed was
recorded. Transects were walked to ground-truth satellite land-use and vegetation type
imagery using Google-Earth imagery, which was found to be satisfactory for the purpose of
assigning land-use and vegetation type categories.
Based on geological/hydrogeological interpretation and field EC/pH measurements, sites
were selected for geophysical surveys using geoelectric soundings in a Schlumberger array.



This investigation aimed at identifying the depth of possible deeper water bearing weathered
or fractured formations.
Selected dug wells were pumped and drawdown and recovery was monitored in order to
estimate aquifer hydraulic conductivity and specific yield, analysed using methods of
Moench (1985) and Barker and Herbert (1989). Tests were repeated in March (dry season)
and October (wet season) of 2015. Well tests were conducted on seven hand dug wells in
Dangila woreda.

*Hydrometric data*
Time series data were available from the national hydrometric network for the Kilti river
gauge at Durbete (Figure 1), and for rainfall and potential evapotranspiration from a
meteorological station near Dangila town. A 7-year period of daily data from January 1997 to
December 2003 was chosen for which almost complete data were available. The daily rainfall
amounts were compared against data from the Tropical Rainfall Monitoring Mission
(TRMM), to determine if they are likely to be representative of the spatial average over the
catchment area.
The river flow data were processed to identify baseflow using a standard flow separation
method (Tallaksen and van Lanen, 2004 ). Various other methods exist for flow separation,
but this provided a consistent approach to estimate the seasonal contribution from
groundwater to the river flow during years with different meteorological conditions.

*Community-based mapping and monitoring*
Following selection of the Dangeshta kebele (sub-district) as the focus site, gender-separated
focus groups were arranged with a Dangila woreda official. These involved firstly a
participatory mapping exercise of available local water resources and areas of land used for
pastoral and crop agriculture, followed by a broader discussion of existing understanding of
the hydrological system, current water use, and constraints and aspirations for agricultural
development. Subsequently, a small sub-group of the participants assisted in identifying
appropriate sites on two of the main river systems for monitoring river levels, as well as sites
for monitoring rainfall and groundwater levels. Two standard river staff gauges were installed
by the community, a suitable site was identified for installation of a non-recoding (manual)
raingauge and 5 shallow hand-dug wells were selected to be monitored using a dipmeter.



These activities were carried out by members of the community, from whom observers were
selected by the community to take daily readings. A workshop was then held to demonstrate
the equipment and its use to a mixed gender and age group audience. The installations and
training were carried out in February 2014, and daily monitoring has continued without
interruption and is still continuing up to and beyond the time of writing (November 2015).
This close engagement with the community has ensured that the equipment has been
protected as there is a sense of ownership by the community. Initial information arising from
the monitoring has been fed back to the communities with the aim of demonstrating the
usefulness of this level of quantitative understanding in order to ensure there is motivation for
continued monitoring.

4.2    Results of resource assessment
*Hydrogeological assessments*
Water-table depth is controlled by topography and geology with clear seasonal variations.
Near the end of the dry season in March/April within the floodplains, where the solid geology
is at a depth of around 4 m, the water-table lies at around 2 m. The water-table can often be
seen as a seepage face at this depth within river bank sections in alluvial sediment. However,
on the larger and steeper slopes where rock-head is around 15 m deep the water-table is at a
depth of around 12 m.

Despite the shallow aquifer being considered to be the weathered basalt regolith and alluvial
materials above the solid geology, it is possible that fractures within the solid geology are
influential to the hydrogeological regime. The geophysical surveys indicated that the
maximum depth of the weathered layer is around 30m, and that fractured zones may exist to
depths of 100-200m. Heterogeneities within the regolith, such as the clay content and the
fractured or vesicular nature of the pre-weathered rock, determine the productivity of a well,
though this is very difficult to estimate prior to excavation. Fissure flow in the deeper zones
is likely to be very restricted, as any fractures are probably filled with weathered material
with the same properties as the overlying materials.

From available geological mapping, four hydrogeological zones were initially identified
within Dangila woreda (Figure 2), which were defined by reclassification of an existing
geological map on the basis of their potential to support small-scale irrigation as follows:



Zone 1: High potential
Loamy soil underlain by sandy clay to depth of up to 4m. Regolith layer reaches 1.5m thick.
Localised pyroclastic fan deposits. High probability of well yield > 1 l/s.
Zone 2: Good potential
Alluvial material 1-2m thick underlain by sandy clay layer up to 3m thick. Regolith layer
reaches 1.5m thick. Weathered basalt with brown, grey and dark brown altered layers up to
25m thick. Good probability of well yield > 1l/s.
Zone 3: Moderate potential
Alluvial material 1 - 2m thick underlain by sandy clay layer 1 - 4m thick. Regolith layer 0.5 -
1.2m thick. Weathered Tertiary basalt up to 16m thick. High risk of well yield < 1 l/s.
Zone 4: Low potential
Sandy to silty clay soil 0.5 – 5.0m deep. Underlain by fresh to slightly weathered dominantly
massive trachyte of variable thickness. Very unlikely to achieve well yield > 1 l/s.

**Figure 2 here**

Following initial reconnaissance surveys and community workshops, it became evident that
topography has a significant influence on borehole locations and most likely also on well
yields. Lowland areas comprising expansive floodplains and low relief topography are
considered to be of high potential for productive groundwater use.  A second map of
groundwater potentials was therefore produced based on surface topography (Figure 3), with
areas being defined by visual interpretation of satellite imagery. Comparison between Figures
2 and 3 shows broad similarities between the low groundwater potential zones in each map
which are generally located on higher ground near the catchment boundaries and along the
divide between the two main drainage areas within the woreda, and between the very high
potential zone in the geology-based map (Figure 2) and the high potential zone along the
valley draining to the south-west in the topography-based map (Figure 3). However, our
surveys, supported by further evidence given below, confirmed the importance of
topographic controls, so the other valley floors to the north-east of the topographic-based map
are also considered to be of relatively high potential (Figure 3).

**Figure 3 here**






Well tests were conducted on seven hand dug wells in Dangila woreda. Both the pumping
and the recovery data were analysed and provided consistent results confirming the suitability
of the methods. Hydraulic conductivity estimates ranged from 0.27 to 5.78 m/d in the dry
season and from 0.93 to 22.3 m/day in the wet season, which are typical values for weathered
basalt regolith. Specific yield estimations have a wider range and are more uncertain though
the mean value is as would be expected. A summary of the results is presented in Table 4.
They confirm that well yields of 1 l/s are achievable.

**Table 4 here**

The locations of the five wells and the raingauge monitored by the Dangeshta community are
shown in Figure 4, against the background of a Google satellite image. It is clearly evident
that these wells follow the general pattern of being mostly close to the edge of the
floodplains, where they remain accessible for the whole year, but are downslope from the
higher ground which provides recharge.

**Figure 4 here**

Records of groundwater levels and rainfall monitored by the local community for the period
April 2014 to November 2015 are shown in Figure 5. These show that only one of the wells
(MW1) dries out completely early in the dry season of 2014-15. Three of the wells (MW2,
MW3 and MW5) show similar behaviour, draining exponentially through most of the dry
season, with small but non-zero depths of water present throughout the season. Water depths
in well MW4 remain high through most of the dry season, before falling sharply in April/May
(there was a period of missing data during this time, but a similar pattern was observed in the
same months of the previous year). These data do, however, show that all the wells
maintained usable water levels into at least the end of December, and in some cases for
considerably longer.

**Figure 5 here**



*Hydrological assessments*

An assessment of the hydrology of the Kilti catchment (Figure 1) for the period 1997-2003 provides insights into groundwater availability within the wider catchment area. Rainfall data for this period was compared with a longer record (1993 – 2014) of monthly rainfall in order to allow an assessment of whether it reflected a sufficiently wide range of conditions. It was found that 1999, 2000 and 1997 represent wet years (96% , 86% and 73% probability of non-exceedance respectively), while 2002 and 2003 represent dry years (9% and 14% probability of non-exceedance respectively) and 1998 represents an average year (40% probability of non-exceedance). The data for 1997-2003 therefore provide an adequate representation of longer term variability.

Annual water balance components for the Kilti catchment are summarised in Table 5 and shown in Figure 6. The catchment receives about 1600 mm/year of rainfall, of which about 200 mm/year enters the groundwater as recharge, discharging to the river as baseflow and with a similar amount of rapid runoff contributing to a total river flow of about 400 mm/year. It can be seen that the wettest year (rainfall 1960 mm) yields 12.8% baseflow, whereas the driest year (rainfall 1350 mm) yields 15.8% baseflow. The lowest value of baseflow is 82% of the mean baseflow which suggests a degree of buffering and indicates that groundwater is available even in a very dry year.

**Table 5 here**

**Figure 6 here**

Mean monthly water balance components for the period 1997-2003 are summarised in Table 6 and shown in Figure 7. The shape of the annual Kilti hydrograph follows that of the annual precipitation cycle. It can be seen that baseflow does not begin to recover until June, thus indicating that groundwater recharge during Belg season (early 'small' wet season) is minimal. However, there is evidence of baseflow persistence beyond the cessation of Kiremt season (main wet season). Mean baseflows for 1997-2003 at the end of the months of September to December are estimated as 8.8, 5.3, 2.1 and 0.93 m$^3$/s respectively, following an exponential decline indicative of natural drainage of groundwater within the catchment. During the driest year of 2002 with rainfall non-exceedance of only 9% based on the long-





term data, the baseflow at the end of December remained at 0.52 m$^3$/s representing 43% of
the mean value for that date, indicating that groundwater remains available at this time even
during dry years.

**Table 6 here**

**Figure 7 here**


**5.    Discussion**
In the past few decades in Asia, a paradigm shift has occurred in irrigation practice, such that
distributed irrigation using privately owned wells and small motorised pumps has expanded
rapidly. This development has enabled smallholder farmers to diversify their farming systems
and grow high-value crops for the market, thus bringing livelihood benefits whilst posing
challenges of resource management and governance. There is growing, but patchy, evidence
that a similar 'irrigation revolution' is happening in SSA (Dessalegn and Merrey, 2015).
There is an expanding literature on smallholder groundwater irrigation in SSA (Giordano,
2006; Giordano and Villholth, 2007; Siebert et al, 2010; Pavelic et al, 2013; Villholth, 2013;
Altchenko and Villholth, 2015). The focus has generally been on assessing potential at
country level and, as identified by Dessalegn and Merrey (2015), there is a need for these
broad evaluations to be supplemented by "localised and detailed assessments". The case
study presented here for Dangila woreda in Ethiopia is an attempt to deliver such an
assessment. It would be unwise to generalise from this case study to the whole of SSA, but as
with the study of Fogera woreda, presented by Dessalegn and Merrey (2015), useful insights
into the wider issues are revealed by the localised case study approach.
This detailed case study has explored the feasibility of exploiting shallow groundwater for
small-scale irrigation over a range of rainfall conditions. Variability of rainfall (9% to 96%
probability of non-exceedance) does not translate into equivalent variability in groundwater
levels and baseflow. Groundwater levels observed in most shallow wells persist into the dry
season to at least the end of December, indication that water is potentially available for
irrigation use during the period after the cessation of the wet season (typically mid Oct).



Catchment baseflows also persist to at least the end of December, even during dry years,
indicating that groundwater is available more widely across the catchment during this period.
Well tests indicate that shallow wells (< 20m) can support abstraction rates of 3.6 m$^3$/hr,
which are sufficient to support small-scale irrigation, at the end of the Kiremt wet season
from October to December. A single well can support irrigated cropping on a plot up to 1ha
provided that crops are planted sufficiently early to make use of rainfall in the later part of the
Kiremt season, and avoid the second part of the dry season when groundwater levels have
generally declined through natural drainage, and which may be required to support other
environmental requirements.
Understanding the resource is necessary, but not sufficient, to guarantee sustainable
management. Small-scale irrigation from shallow groundwater, like any other type of
irrigation development, should be seen as a socio-technical problem (Dessalegn and Merrey,
2015). This implies that the social dimensions of irrigation are as important as the technical
dimensions. Social dimensions include issues of governance such as organisation of water
use, collective action and conflict resolution. Technical dimensions in this case include issues
of resource assessment and water lifting. Sustainable development of small-scale irrigation
from shallow groundwater in SSA will require support from external agents (hydrogeologists,
irrigation experts etc), but most importantly will depend upon a devolved participatory
approach to local resource management at the community level. In the case of Ethiopia, the
woreda (ie. district) is the appropriate scale in that it provides the interface between local
communities and external agents.
There is a need for further action-research at this scale in places like Dangila woreda to
develop capacity for the state to function in a dual role as facilitator and regulator of GWI.
Community based monitoring (citizen science) has been shown to be valuable in providing
the data required for resource management while also providing an entry-point for external
agents. There is a case for investigating the feasibility of establishing a cadre of local 'para-
hydrologists' to act as intermediaries between local communities and external agents in the
long term. Para-hydrologists are expected help ensure quality of community-led monitoring
data as well as to play a key role in facilitating bi-directional information exchange between
technical professionals and community members. Experience in India (Shah, 2007) has
demonstrated the value of providing an appropriate level of technical training in hydrology
and hydrogeology in promoting community level groundwater governance.





**6.    Conclusion**
Shallow groundwater resources represent a neglected opportunity for sustainable
intensification of small-scale agriculture in SSA. Concerns over low aquifer transmissivity,
low well yields, aquifer vulnerability and resource conflict are exaggerated. Shallow
groundwater (< 20m depth) is accessible to small-scale farmers and should be seen as a
strategic resource.  There is a need to develop capacity for the state to function in a dual role
as facilitator and regulator of GWI. However, the localised nature of shallow aquifers will
require an approach based around participatory resource management by local communities.
There is widespread use of shallow groundwater for domestic supply in most SSA countries,
and indigenous knowledge generally exists on the seasonal performance of wells during
typical and drought years. This knowledge is localised, qualitative and unrecorded, but it
provides an entry-point for a participatory approach.
We propose an approach to developing irrigation from shallow groundwater in SSA with a
focus on community-led adaptive resource management. This is based on two main premises:
•       that a 'bottom-up' approach with close engagement between local communities and
professionals is necessary for development of shallow groundwater resources for small scale
irrigation;
•       that an adaptive approach to integrated management of groundwater and surface water
resources is necessary for long-term sustainability, and this requires quantitative hydrological
monitoring at the local scale, particularly of groundwater levels.

**Acknowledgements**
This work was funded by the NERC/DfID UpGro programme under Grant NE/L002019/1.
We are grateful for the co-operation of many people and organisations in Ethiopia,
particularly the local communities in the Dangila woreda. Further details of the study are
available at http://research.ncl.ac.uk/amgraf.




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

biophysical and socioeconomic approach. IFPRI Discussion Paper 993. Washington DC:
International Food Policy Research Institute (IFPRI).





**Figure and table captions**


Figure 1.    Case study site: Dangila woreda in Amhara region, Ethiopia
Figure 2.    Groundwater potential zones, based on reclassification of geological map
Figure 3.    Groundwater potential zones, based on topographic analysis
Figure 4.    Locations of community monitoring wells and rain gauge
Figure 5.    Daily community observed data for 2014-15: groundwater levels are plotted
relative to base of well to show water column depth (well depths are: MW1 6.00m; MW2
6.89m; MW3 4.18m; MW4 9.17m; MW5 8.44m)
Figure 6.    Annual river discharge and baseflow for the Kilti catchment (1997-2003)
Figure 7.    Mean monthly river discharge and baseflow for the Kilti catchment (1997-2003)



Table 1: Pumping rate for human-powered device operating at 50% efficiency
Table 2: Pumping rate for animal-powered device operating at 50% efficiency
Table 3: Daily water use (m3/day) under a range of irrigation demands at 80% efficiency
Table 4: Aquifer properties determined by well tests using methods of Moench (1985) and
Barker and Herbert (1989): hydraulic conductivity (K); specific yield (SY)
Table 5: Annual water balance data for 1997-2003 (mm)
Table 6: Mean monthly water balance data for 1997-2003 (mm/day)




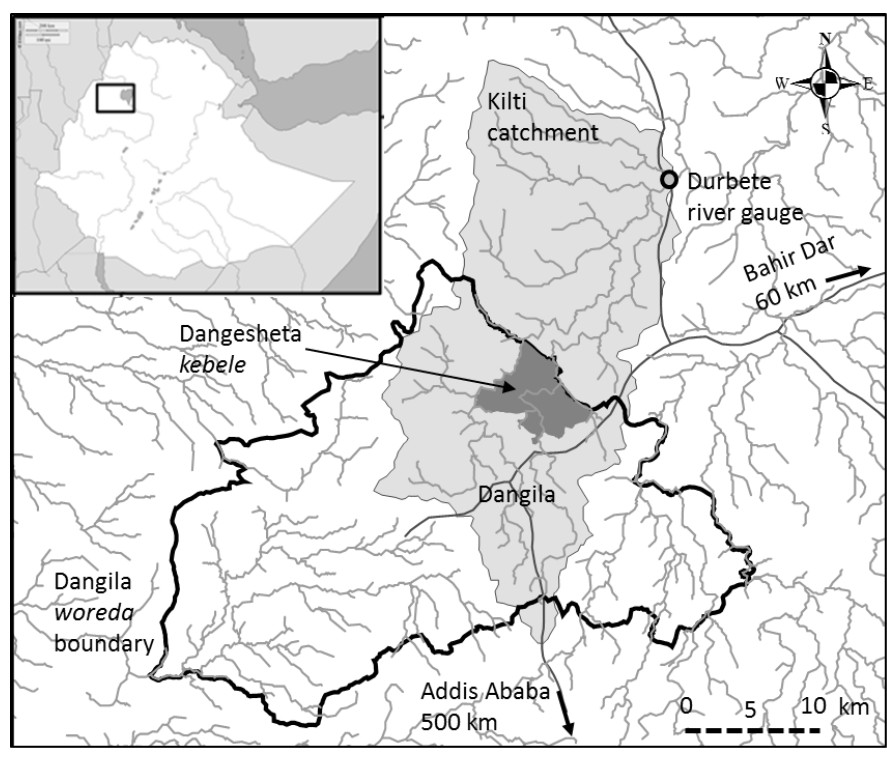


Figure 1. Case study site: Dangila woreda in Amhara region, Ethiopia




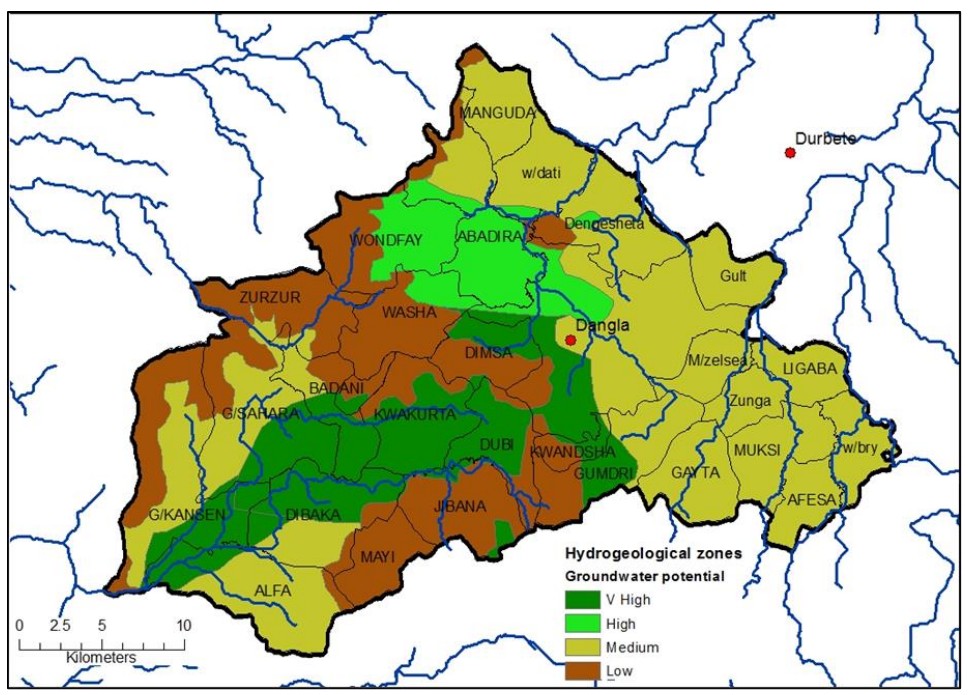


Figure 2.  Groundwater potential zones, based on reclassification of geological map






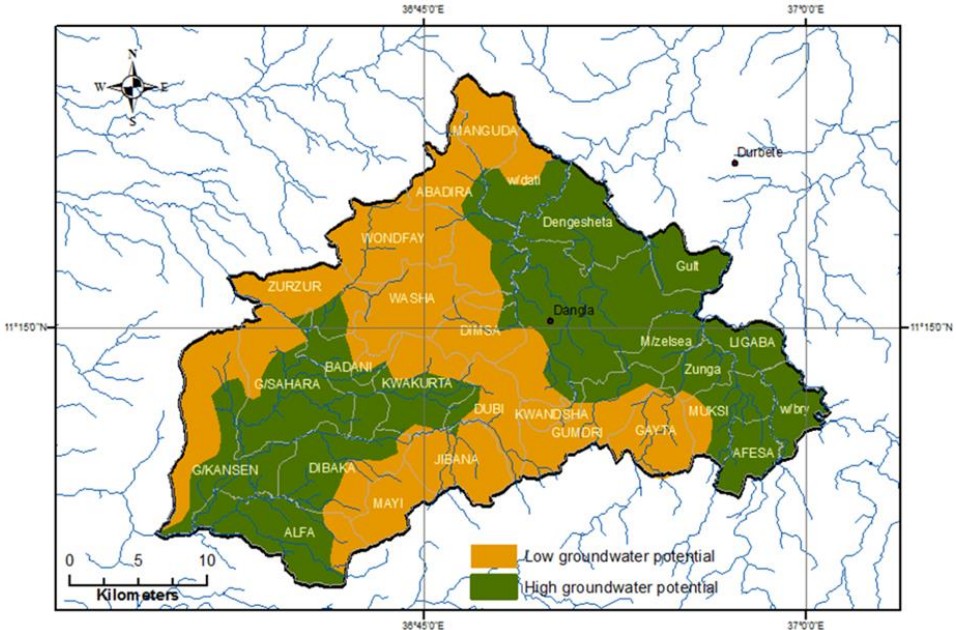


Figure 3. Groundwater potential zones, based on topographic analysis





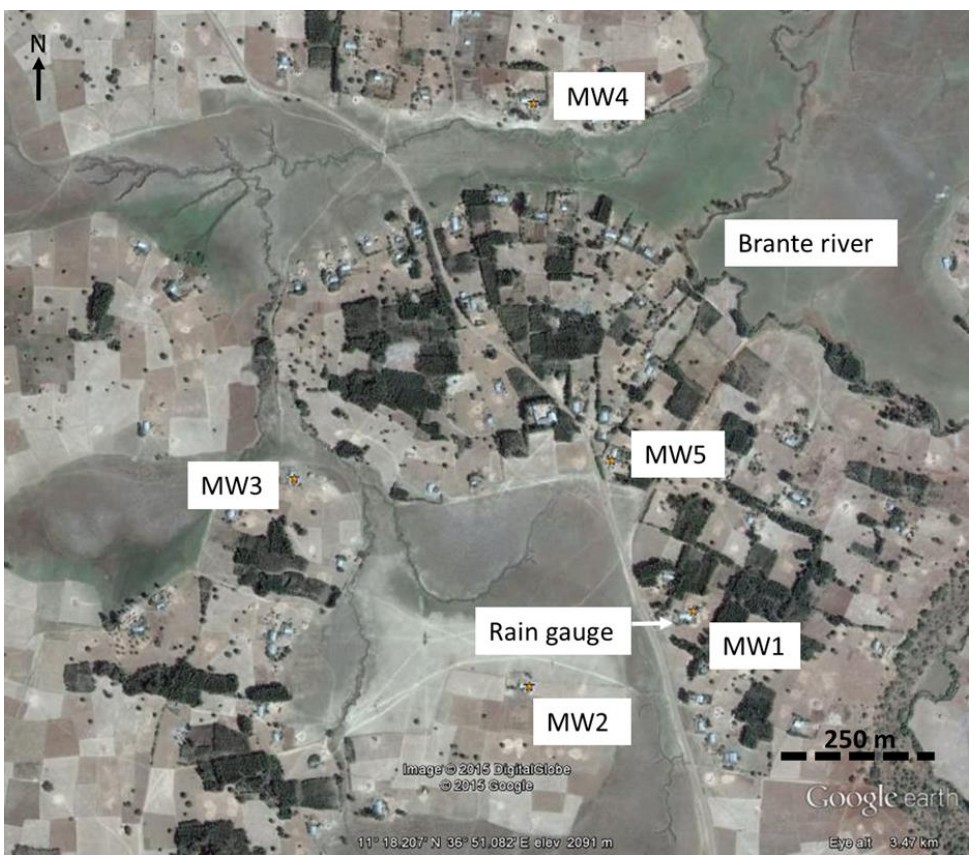


Figure 4.  Locations of community monitoring wells and rain gauge






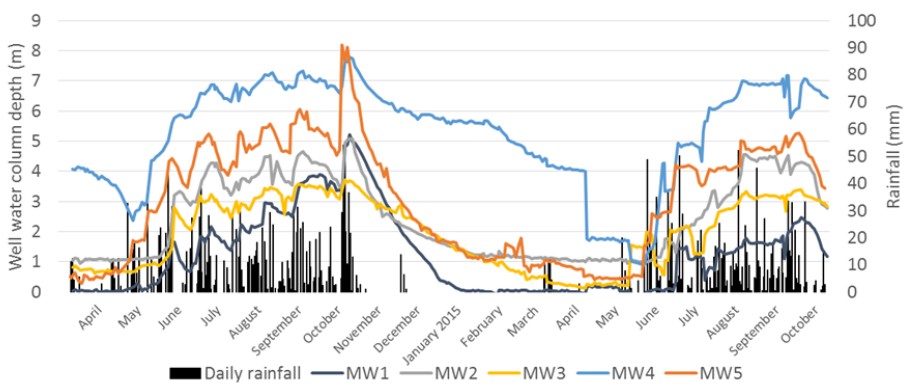

Figure 5: Daily community observed data for 2014-15: groundwater levels are plotted relative
to base of well to show water column depth (well depths are: MW1 6.00m; MW2 6.89m;
MW3 4.18m; MW4 9.17m; MW5 8.44m)





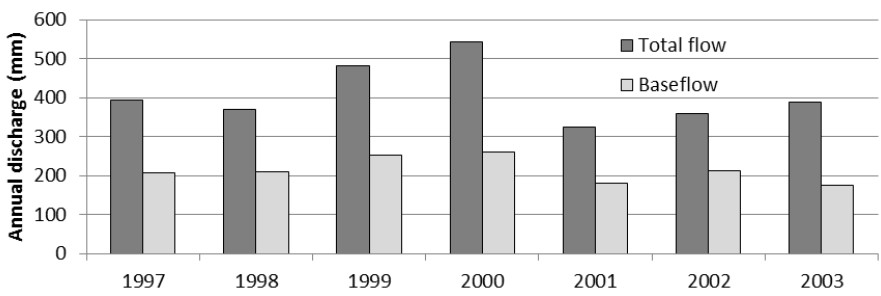

Figure 6. Annual river discharge and baseflow for the Kilti catchment (1997-2003)

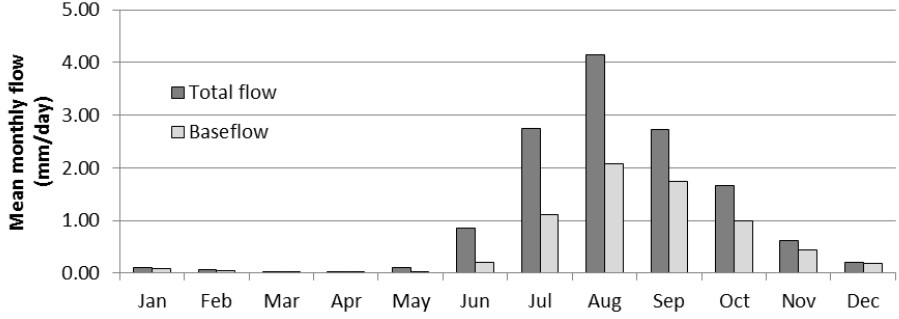

Figure 7. Mean monthly river discharge and baseflow for the Kilti catchment (1997-2003)





| Head (m) | 0.5 | 1.0 | 2.5 | 5.0 | 10.0 | 20.0 |
|----------|-----|-----|-----|-----|------|------|
| Rate (m³/h) | 27.5 | 13.8 | 5.5 | 2.7 | 1.3 | 0.7 |


Table 1: Pumping rate for human-powered device operating at 50% efficiency



| Animal | Weight (kg) | Power (W) | Pumping rate (m³/h) at various heads | | | |
|--------|-------------|-----------|------|------|-------|-------|
| | | | 1.0m | 5.0m | 10.0m | 20.0m |
| Mule | 350 - 500 | 300 - 600 | 54 - 108 | 10.8 - 21.6 | 5.4 - 10.8 | 2.7 – 5.4 |
| Donkey | 150 - 300 | 75 - 200 | 13.8 - 36.8 | 2.7 - 7.2 | 1.3 - 3.7 | 0.7 - 1.8 |
| Bullock/ox | 500 - 900 | 300 - 500 | 54 - 90 | 10.8 - 18.0 | 5.4 - 9.0 | 2.7 – 4.5 |


Table 2: Pumping rate for animal-powered device operating at 50% efficiency


| Irrigation demand (mm/day) | Area irrigated (ha) | | | |
|------|------|------|------|------|
| | 0.1 | 0.25 | 0.5 | 1.0 |
| 5 | 4.0 | 10.0 | 20.0 | 40.0 |
| 6 | 4.8 | 12.0 | 24.0 | 48.0 |
| 7 | 5.6 | 14.0 | 28.0 | 56.0 |
| 8 | 6.4 | 16.0 | 32.0 | 64.0 |


Table 3: Daily water use (m³/day) under a range of irrigation demands at 80% efficiency





|  | Dry season | | Wet season | |
| --- | --- | --- | --- | --- |
|  | K (m/day) | $S_Y$ | K (m/day) | $S_Y$ |
| Mean | 2.10 | 0.097 | 8.79 | 0.074 |
| Median | 1.32 | 0.075 | 6.20 | 0.054 |
| St Dev | 1.84 | 0.095 | 7.52 | 0.112 |


Table 4. Aquifer properties determined by well tests using methods of Moench (1985) and
Barker and Herbert (1989): hydraulic conductivity (K); specific yield ($S_Y$)



|  | 1997 | 1998 | 1999 | 2000 | 2001 | 2002 | 2003 | Mean |
| --- | --- | --- | --- | --- | --- | --- | --- | --- |
| Rainfall | 1667 | 1555 | 1959 | 1896 | 1411 | 1350 | 1369 | 1601 |
| Potential Evapotranspiration | 1451 | 1425 | 1417 | 1416 | 1405 | 1415 | 1422 | 1422 |
| Discharge | 395 | 368 | 481 | 544 | 324 | 358 | 388 | 408 |
| Baseflow | 208 | 210 | 252 | 259 | 179 | 213 | 175 | 214 |


Table 5.  Annual water balance data for 1997-2003 (mm)




|  | J | F | M | A | M | J | J | A | S | O | N | D | Total |
| --- | --- | --- | --- | --- | --- | --- | --- | --- | --- | --- | --- | --- | --- |
| Rainfall | 0.01 | 0.06 | 0.46 | 1.00 | 4.76 | 8.38 | 10.84 | 11.06 | 7.79 | 5.02 | 1.39 | 0.07 | 4.26 |
| Evaporation | 3.51 | 3.99 | 4.42 | 4.75 | 4.42 | 3.99 | 3.42 | 3.27 | 3.73 | 4.02 | 3.80 | 3.42 | 3.89 |
| Discharge | 0.11 | 0.06 | 0.03 | 0.03 | 0.12 | 0.85 | 2.75 | 4.14 | 2.73 | 1.67 | 0.61 | 0.20 | 1.12 |
| Baseflow | 0.09 | 0.05 | 0.02 | 0.01 | 0.03 | 0.21 | 1.12 | 2.08 | 1.74 | 1.00 | 0.44 | 0.18 | 0.58 |


Table 6.   Mean monthly water balance data for 1997-2003 (mm/day)

