# Peer review of "Shallow groundwater in sub-Saharan Africa: neglected opportunity"

_Hydrology and Earth System Sciences, 2015_

## Short Comment (SC1) · 21 Jan 2016

Having worked throughout Africa, Middle East and Latin America for 30 years, it was my impression that the use of shallow groundwater for irrigation was taken for granted as I have seen it everywhere accessed and used for this purpose. And, electrical resistivity is probably not the best method for location of hand dug wells or boreholes. It dos not measure any proprty of the watr but rather the resistivity of the aquifer rock material which has been correlated with groundwater occurrence. The only geophysical groundwater exploration method is Thermonics (www.agwconsultants.com) that has been used in Ecuador, Pakistan, and throughout the United States. On a project for the Government of Burkina Faso some years ago, I was retained to teach Burkinabe
[Figure]

Hydrogeologists how to use fracture trace analysis. Fracture trace analysis is sometimes called lineament analysis. As the authors indicate 40% of Africa is underlain by crystalline rock where water occurs commonly in fractures within the rock. After training was completed we selected several villages to demonstrate the method in the field. Lineament analysis makes use of aerial photographs and the trick is to find on the ground the location found on the aerial photograph. This method was developed in the 1960's by two of my professors at Penn State, Dick Parizak and Larry Lattman. Dick was the hydrogeologist and Larry was the photo-interpreter. At the first village we went to (whose name escapes me at the moment) after meeting with the village leaders we set about locating a well site. Having done that, and as we were leaving the village, a small boy came running up to us and with great excitement he took us to a site picked by GOBF hydrogeologists a week earlier where used electrical resistivity. It was only feet away from our chosen site. Difference was, it took me 10 minutes to locate the same site. Lineament analysis has been used extensively by Swedish AID hydrogeologists in Nigeria and elsewhere and there are many publications in the literature. I have seen hand dug wells used for irrigation in Kenya, ex-Zaire, Tunisia, Mali, Mauritania, Sierra Leone, Somalia, Burkina Faso, and elsewhere. In Mauritania the hand dug wells are generally constructed with the help of primacord and explosives. They are commonly constructed in alluvium of wadiis where the shallow wadi aquifers are recharged from rainfall and flood events. I think each area has pretty much defined the type of ground water occurrence and they have worked out the best extraction methods. A hundred years ago, hand dug wells were constructed in Cyprus. In the Ayia Marina coastal plain of Cyprus. I myself monitored water levels in more than 100 of these small scale shallow wells They chose to use small diesel engines manufactured by Lister and Skoda and others. The most common was the 5 HP Lister with a small flywheel and a long flat canvas belt that powered a centrifugal belt connected to a centrifugal pump situated close to the water table. Small farmers in the area had organized a cooperative and all of their diesel fuel and other supplies were bought though their cooperative. Further, the cooperative marketed their products both into

the local wholesale markets and internationally. R A Lister & Company was founded in Dursley, Gloucestershire, England, in 1867 by Sir Robert Ashton Lister (1845–1929), to produce agricultural machinery. A Google search will provide a wealth of information on he famous Lister cold-start diesel engines.

———————————————

---

## Referee Comment (RC1) · Anonymous Referee #1 · 31 Jan 2016

I agree with the authors that shallow groundwater used for small-scale irrigated agriculture in sub-Saharan Africa has the potential to be an important factor for improving the living conditions of subsistence farmers and, if well managed, can be a sustainable option. It is good to see members of the academic community engaging with work in SSA related to subsistence farming and the technical aspects of water supply. However, I am recommending that this manuscript be rejected from HESS.

I am struggling to see this manuscript as a research article; in my view, it is not sufficiently novel in approach, methods, or results to be considered original research. As a case study, it is detailed, but this is not sufficient.

- The methods and approach are not novel; with the exception of the fact that the au-

thors undertook community-based monitoring to collect data and they used a Schlumberger Array in the resistivity survey (as opposed to dipole) the approach is the same as that which is undertaken for every hydrogeological survey in many countries in SSA (e.g., Kenya, Tanzania, Uganda) - these surveys are done whenever a drilling permit is sought.

- In much of SSA, fluvial unconsolidated aquifers are well known as a consistent source of groundwater in the dry season(s); in fact they are often used for both domestic purposes and agricultural activities in many other SSA countries. What limits their used for irrigated agriculture (in cases where they are as reliable as the one studied by the authors) is not that they are an unknown resources, its that the communities do not have the economic resources to invest in the infrastructure. The pump options mentioned by the authors exist in SSA (as do drilling rigs, hydrogeologists, etc.) but most of these items are out of reach financially by many rural SSA communities; this is an issue of government/NGO resources rather than a knowledge gap.

- The discussion section of this manuscript felt much like a rehash of the intro/literature review. It didn't move the narrative forward. This may be more of a stylistic critique, but much of the intro and discussion felt like an op-ed rather than the basis for original research.

Recommendations

- Incorporate more study locations so that a more generalized understanding of alluvial groundwater availability can be developed

- Pursue the community-based monitoring aspect: if the authors can demonstrate that this can be maintained in SSA for the long-term, it could be published as an alternative method for gather hydrological data in remote areas with little available information

- Incorporate some modeling aspect to the study (some possibilities include: impacts of climatic variability, drought vs flood conditions) so that the results and interpretations

can go beyond where they are right now.

- With some additional data (longer-term monitoring, comparison to surrounding catchments), this may be publishable as a case study in another journal
* * *

---

## Referee Comment (RC2) · Anonymous Referee #2 · 29 Mar 2016

I agree with 'Anonymous Referee #1' that the paper makes a good case for the potential of sustainable shallow groundwater use for small-scale irrigated agriculture in sub-Saharan Africa. However the paper would be improved by extending the case beyond the study site cited. There is a very large body of work by the National University's, CGIAR centres, and NARs throughout SSA on Ag-water options and potential, including shallow GW. Some of this is referenced (de Fraiture/Pavelic/MacDonald) but as much of this work remains in local journals and grey literature/national level reports, I do feel that it is important to bring the 'Northern' literature up-to-date on the issue. I also agree with the referee that the paper is not particularly innovative in methods or approach, but I recommend that the paper should be accepted by HESS with consideration of the following points:

- With regard to recharge and groundwater potential and sustainability in Ethiopia, the study should consider the work of Steenhuis et al / Cornell-Barhidar University teams, on rainfall-runoff in the Ethiopian highlands. (e.g. line 474?)

- The paper should draw more on other case studies regionally, to scale-out their own findings and explore the potential of shallow-GW beyond the one sub-catchment studied.

- There is no/very limited discussion of the cultural / non-technical limitations to the expansion of more formalized shallow-GW development in Ethiopia. Lack of secure tenure and farm size remain a major limitation to any improvement / investment in infrastructure (line 562 mentions typical farm size – it could be useful to indicate that this is often less than 1ha). There is a tradition of GW use for domestic use (over 70% of the population use GW for domestic use – see Macalister/Pavelic in Awulachew et al, 2012), including springs used widely for drinking water and even bottling (see Ambo in ET highlands), and for livestock watering (the famous Borena in the south) but irrigation is typically low intensity and localized. The study did provide a very good example of community engagement and this could be made much more of.

- There is mention of the potential for drip irrigation which has had limited success in SSA outside of large commercial horticulture (including the Foreign Direct Investment projects in the Ethiopian Rift Valley) and low efficiency surface methods still dominate. Any recommendation of more efficient methods, where likely supply rates are low, should also give consideration to the capacity/skill to source and maintain equipment. Certainly in the past, there was a shortage of functional drilling rigs in addition to affordable pumps (cost of pumps is mentioned). And though this is likely to change with the proliferation of more affordable Chinese equipment, there is still a skill/capacity and institutional support gap to be filled to maintain such equipment. Without digressing from the topic too much there is a large body of work on the success and failure factors

of drip irrigation in SSA (eg. Garb and Friedlander), even very recent (2016 – Yami / Venot / Amjath).

- Given the low capacity of manual/draft powered irrigation (where pumps remain a limitation), and that small-holder production in the ET highlands is predominately rain-fed, one recommendation could be that shallow well irrigation could be introduced as supplementary irrigation during dry spells, which are a major threat that is growing more frequent and severe in the changing climate.

- Line 537 mentions the potential for a small irrigation revolution in ET/SSA as a means to poverty reduction through cash crop production, drawing comparison to the Asian experience. Caution should be used when applying an Asian model to agricultural markets in SSA, particularly with poorly connected Ethiopian small holders. Access to market and value chain development is a major constraint on any cash crop potential, and there are many longstanding development projects with volumes of research that highlight this issue (e.g. the ILRI Improving Productivity and Market Success of Ethiopian Farmers program). Mention of cash crop potential should necessarily also refer to these constraints.

---

## Author Comment (AC1) · 24 Apr 2016

We are grateful for the informative comments by Dr Turner. We are not familiar with the Thermonics technique to which he refers. We do not agree with his opinion that "the only geophysical groundwater exploration method is Thermonics". The electrical resistivity method is very well established and is widely used by the Geological Survey of Ethiopia who were our partners in the AMGRAF project. The method is very well suited to investigation of shallow aquifers; fracture zones and the boundary between weathered and unweathered bedrock are easily identified.

We agree with his comment about the utility of lineament analysis particularly for locating higher-yielding boreholes in fracture zones in crystalline bedrock. It is less relevant

for shallow wells where lower yield is acceptable and the regolith is the target rather than deeper rock. In any case it was not our aim to identify well locations, rather we wished to investigate the extent of the shallow aquifer.

We do not agree with his comment that "the use of shallow groundwater for irrigation was taken for granted". We recognise that there are indeed hotspots where this does occur and we report that its use is certainly more extensive than we see in official data on groundwater use. Our intention is to question its neglect by public authorities with the remit to develop and manage natural resources for the benefit of poor people living in rural areas.

One reflection of the official neglect is the failure to make available suitable pumping equipment. We agree that a small diesel engine with a belt drive to a centrifugal pump situated close to the water table has been a successful approach in the past but a long belt drive will be inefficient and problematic. We argue that developing a well is achievable and affordable to 20m depth but available pumping technology is not suitable beyond 10m or less. Electric submersible pumps with photovoltaic power sources are now an attractive alternative.

---

## Author Comment (AC2) · 24 Apr 2016

We are grateful to the anonymous reviewer for his commendation of our efforts in engaging with work in SSA related to subsistence farming and the technical aspects of water supply. We are disappointed that he/she is reluctant to support publication of the experience in HESS.

We share the view of Dessalegn & Merrey (2015), that there is a need for previously published broad evaluations to be supplemented by "localised and detailed assessments". Useful insights into the wider issues are revealed by the location-based case study approach reported in our paper. We have attempted to ensure that its relevance beyond the study site is apparent through our review of the literature. We argue there-

fore that the case study approach is the most appropriate way to explore the potential of and constraints to developing shallow groundwater for small-scale irrigation in SSA.

We further argue that the work is novel in that we adopted a holistic and participatory, research-for-development approach. Our aim in adopting location-based research was to explore the problem from a user's perspective and to deliver a rigorous assessment. This was intended to achieve twin benefits of (i) providing strong evidence to stakeholders within the community of donors and government institutions, and (ii) engaging with local people who represent the primary beneficiaries in a way that both tests and promotes their willingness to participate. The impact pathway was designed into the research from the outset and we considered that there was more to be gained from in-depth research at one case study site rather than attempting to cover greater diversity of sites. Ideally, we would have aimed to achieve both objectives but funding was insufficient. In order to secure additional funding that would allow us to extend the reach we need to establish the credibility of the principles and underlying concepts of our argument. Without wishing to deny its limitations, our aim is to publish the results as they stand in order to argue that we have gathered sufficient evidence to show potential for success.

The reviewer comments that fluvial unconsolidated aquifers are well known as a consistent source of groundwater and that they are often used for both domestic purposes and agricultural activities in many other SSA countries. On the contrary, we reviewed the literature extensively in order to show that these resources suffer from official neglect despite their potential. The groundwater revolution that has transformed irrigation in much of Asia has not been transferred to SSA.

The reviewer further argues that what limits their use for irrigated agriculture is not that they are an unknown resources, it is that the communities do not have the economic resources to invest in the infrastructure. The existence of hotspots of autonomous development of shallow groundwater development indicates that this is a false assumption. We argue that shallow groundwater is the entry point for poor small scale farmers

to intensify production and switch from subsistence to market-oriented agriculture. The problem is official neglect which fails to make available the appropriate technology for well drilling and water lifting. We argue that the nature of the problem is such that a participatory community-based approach is the only way to proceed.

The reviewer offers suggestions for further work on the community-based monitoring aspect, and modelling impacts of climatic variability, drought and flood conditions. These are sound suggestions and we are already pursuing further research along these lines. Publication of our findings to date will help to establish validity of our approach, to allow us to obtain funding to maintain and extend our initial studies at the present and other field sites.

---

## Author Comment (AC3) · 24 Apr 2016

We are grateful to the anonymous reviewer for his/her support for publication of the AMGRAF experience in HESS.

We have already presented our argument above in our response to Reviewer #1 about the case-study focus of our paper. Reviewer #2 makes a valid point that sources in the grey literature could usefully extend what he/she calls 'Northern' literature, but we were restricted by the word limit for the journal. We note that reviewer #1 has expressed some concern that the existing review of literature is rather long and we would therefore welcome advice from the editor on how to proceed.

The reviewer advises that the paper should consider the work of "Steenhuis et al / Cornell-Barhidar" (Bahir-Dar) University teams, on rainfall-runoff in the Ethiopian highlands. We are aware of this literature, and in fact some of the papers include our project colleagues and one of the co-authors of this paper as co-authors. These papers address general process understanding and modelling approaches for rainfall-runoff in the Ethiopian highlands, and address related issues of climate impacts, surface water runoff and sediment erosion, with some community-based approaches for water management, focussed on household water supply. They do not include any specific emphasis on groundwater resources, the focus of our paper, but we do appreciate that they are relevant to our general direction of research in Ethiopia and may be usefully included within our review.

The reviewer comments that there is limited discussion of the cultural / non-technical limitations to the expansion of more formalized shallow-GW development in Ethiopia, e.g. lack of secure tenure and farm size. While we acknowledge these issues ("social dimensions of irrigation are as important as the technical dimensions", lines 566/567), our focus is on emphasising the potential for development, and we argue that "it is feasible to irrigate up to 1 hectare from a single well pumping from 20m deep", lines 332/333), which indicates that even small farms of less than 1ha can benefit from shallow groundwater resources.

There is a tradition of GW use for domestic use (over 70% of the population use GW for domestic use – see Macalister/Pavelic in Awulachew et al, 2012), including springs used widely for drinking water and even bottling (see Ambo in ET highlands), and for livestock watering (the famous Borena in the south) but irrigation is typically low intensity and localized. This latter comment supports our view that reviewer #1 overstates the extent of existing GW irrigation in SSA. We acknowledge concerns over potential conflicts with domestic uses. We do not regard bottling as an issue as this use involves deeper aquifers.

The reviewer comments that the study did provide a very good example of community

engagement and this could be made much more of. We are continuing our research in this area, and could readily emphasise this aspect further in this paper. See our summary comments below.

The reviewer mentions the potential for drip irrigation which has had limited success in SSA outside of large commercial horticulture (including the Foreign Direct Investment projects in the Ethiopian Rift Valley) and notes that low efficiency surface methods still dominate. We recognise that any recommendation of more efficient methods, where likely supply rates are low, should also give consideration to the capacity/skill to source and maintain equipment. As reviewer #2 notes, there is a large body of work on the success and failure factors of drip irrigation in SSA.

The reviewer suggests that one recommendation could be that shallow well irrigation could be introduced as supplementary irrigation during dry spells, which are a major threat that is growing more frequent and severe in the changing climate. We fully agree with this point, and further research is planned to address this directly based on our community-monitored observations combined with modelling and scenario analyses. We have presented and discussed the evidence supporting this throughout the paper, for example see our comment related to baseflow assessment "that a degree of buffering and indicates that groundwater is available even in a very dry year" (lines 508/509), and "A single well can support irrigated cropping on a plot up to 1ha provided that crops are planted sufficiently early to make use of rainfall in the later part of the Kiremt season, and avoid the second part of the dry season when groundwater levels have generally declined through natural drainage, and which may be required to support other environmental requirements" (lines 558-562). Any attempt to promote small-scale intensification must offer risk-reduction strategies allowing for vulnerability to climatic variability, and supplementary irrigation is an attractive option which we aim to demonstrate is viable with evidence from our study site.

The reviewer recommends that mention of cash crop potential should necessarily also refer to market access constraints. We agree with this point. We note that our choice

of general study area was based on "priority given to agricultural transformation in the area and availability of hydrogeological data" (lines 169/170), but that we do not provide detail of how we selected the Dangeshta kebele for the detailed study. We did conduct a detailed preliminary review prior to site selection (line 384), with one of the key criteria being access to a potential market, and it would be possible to indicate this more clearly in the paper.

---

## Author Comment (AC4) · 24 Apr 2016

We are grateful for the comments on our paper, which we will take into account in strengthening our arguments and clarifying some specific issues in our paper. We summarise our responses to some key common comments here.

The reviewers' comments in relation to the literature reviews were not consistent, recommending separately more concise and more extensive reviews, and we would appreciate editorial advice on this. We recognise that significant work has previously been carried out and published on groundwater in Africa, and we have covered the aspects most relevant to our own focus in our literature review. We appreciate the further suggestions made, but we feel that our existing coverage of the literature is sufficient

and appropriate for a journal paper that is not intended to be a review paper.

Both reviewers mentioned the use of other case study sites. We consider that the arguments presented in our paper are properly evidenced from a detailed assessment at one site, supported by published research at other sites within our literature review. We have referenced recent publications indicating the need for localised field studies, and we hope that this paper will encourage wider discussion of the issues, enable subsequent research to investigate other sites, and help in advocating more official recognition of the potential use of shallow groundwater resources.

A particularly novelty in the paper, recognised by both reviewers, is the use of community-based monitoring to support resource management, which is a focus that we continue to pursue and develop further. There is increasing interest in the published literature of the value of alternative sources of knowledge and information, and we have aimed in this paper to add a new perspective by presenting an argument for its necessary role specifically in the context of the use of shallow groundwater for small-scale agricultural use, which we consider is a useful and novel addition to debates in the published literature.

In view of the fact that there has previously been valuable discussion of groundwater potential in SSA within the pages of HESS, but that this work has not identified the potential of the shallow groundwater resource, we believe that our results to-date merit publication in HESS. The paper was shared with Knowledge Broker (Sean Furey) for NERC-DFID research programme 'Unlocking the Potential of Groundwater for the Poor', who commended its publication as follows: "Great paper! There is useful stuff here for the discussions and work on-going about drilling and pumps, but I'm going to use it as an excuse to engage with the agricultural networks, like FARA, as well. I'll keep an eye on the paper for when it goes live. I've posted the provisional details here: http://upgro.org/publications-papers/peer-reviewed-journal-papers-open-access/. And in the next couple of days I'll work up a short news item for the website and to circulate more widely for when the paper is definitively published."